# Nutritionally Enriched Muffins from Roselle Calyx Extract Using Response Surface Methodology

**DOI:** 10.3390/foods11243982

**Published:** 2022-12-08

**Authors:** Sengnolotha Marak, Nutan Kaushik, Alexander Dikiy, Elena Shumilina, Eva Falch

**Affiliations:** 1Amity Food and Agriculture Foundation, Amity University Uttar Pradesh, Noida 201313, UP, India; 2Amity Institute of Food Technology, Amity University Uttar Pradesh, Noida 201313, UP, India; 3Department of Biotechnology and Food Sciences, Norwegian University of Science and Technology (NTNU), 7491 Trondheim, Norway

**Keywords:** *Hibiscus sabdariffa*, Roselle calyx extract, Roselle muffin, Response Surface Methodology

## Abstract

*Hibiscus sabdariffa*, often called Roselle, is a flowering plant with a variety of traditional medicinal uses. Its calyx, with a bright and attractive red color, produces a tart and pleasant acidic taste. The purpose of this study was to develop a Roselle muffin and assess the acceptability, nutrition, and shelf life of the muffin using its ingredients. The muffin was developed using different formulations in different proportions resulting from Response Surface Methodology (RSM). Sensory parameters were used to assess the muffin’s acceptability. According to the findings, the combination of extract volume 45.37 mL, citric acid 1.11 g, and sodium bicarbonate 1.67 g produces the best muffin, with the panelist’s sensory scores reaching up to 84%. The outcome of the study suggests muffins baked with the Roselle calyx extract have high antioxidant (12.53 ± 0.13)%, anthocyanin (126.63 ± 1.96) mg Cyn-3-glu/100 g, phenolic (12.91 ± 0.69) mg GAE/100 g, and ascorbic acid (12.10 ± 0.89) mg/100 g contents. The microbial shelf life of the developed muffin is estimated to be 6 days at room temperature. The study findings can therefore be utilized in the development of foods containing Roselle calyx extract.

## 1. Introduction

Due to their bioactive components, adding plant extracts to food products has become more popular. Natural bioactive components have a significant impact on daily activities and are linked to numerous health advantages and minimal toxicity. Many diseases have been treated with plant-derived medicines because of their long-recognized therapeutic properties [1]. In the development of functional foods and the treatment of human diseases, several active substances with various biological effects are widely used [2,3,4]. An excellent source of molecules for the development of nutraceuticals, functional foods, and food additives are offered by natural bioactive compounds with a wide variety of structural and functional properties [5].

The aqueous extract of *H. sabdariffa* calyces contains a wide spectrum of several groups of bioactive compounds, including anthocyanins, polysaccharides, flavonoids, and other organic compounds including betaine and hibiscus acid [6]. Anthocyanins, which are abundant in calyces and are responsible for their red pigmentation, have undergone extensive scientific research owing to their distinctive color characteristics [7,8]. Numerous studies have demonstrated that *H. sabdariffa* and its extract exhibit useful qualities and water-soluble bioactive components that can be exploited to develop new products with additional nutritional attributes that might enhance consumers’ health [6,9,10]. Consumer acceptability of any dish is greatly influenced by its color. Natural colors have become more prominent due to a growing public concern over the toxicity of synthetic colorants [11]. The calyces of *H. sabdariffa* are thought to be excellent and attractive sources of red colorants that are water-soluble and could be used as natural food colorings [12,13]. In parts of the world including West Africa and South Asia, *H. sabdariffa* is used to make tea. It has a tart flavor, and it is sold commercially in these regions as jams and juices [7].

While developing any bakery products, quality parameters such as texture, phytochemical content, and shelf life are important to consider. Muffins that are well-aerated, elastic, and possess a degree of springiness are high-quality and fresh [14]. Another important texture parameter is chewiness, which is mostly related to how tough it is to chew before swallowing [15].

In addition to offering basic nutrition, phytochemicals present in fruits, vegetables, grains, and other plant foods may also provide health advantages by lowering the risk of serious chronic diseases [16]. However, during processing, these phytochemicals are subjected to a variety of degradations. The physical characteristics and chemical composition of foods change because of cooking, which makes this alteration quite evident [17]. Due to its processing conditions, the results on the changes in phytochemicals during household cooking were inconsistent and occasionally conflicting. For example, Blessington et al. found that cooking potatoes in various ways—including boiling, baking, frying, and microwaving—significantly raises their overall phenolic content and antioxidant activity [18]. On the contrary, Xu et al. discovered that boiling, baking, and microwaving decreased phytochemical concentrations and their antioxidant activity [19].

Shelf life is a particularly important aspect in the production of food products. Low- and medium-moisture bakery items are more prone to physical and chemical deterioration, whereas high-moisture products are more vulnerable to microbiological decomposition by bacteria, yeast, and molds. Although the heat while baking destroys mold and mold spores by thermal reactivity, post-baking contamination from airborne mold spores occurs during handling procedures such as cooling and packing [20]. According to Gilbert et al., cakes with aerobic plate counts greater than 6 log10 CFU had unacceptable microbiological quality, whereas acceptable values ranged from 5 log10 CFU to 6 log10 CFU [21]. The generic microbiological standard for cakes and pastries, which was published by the Institute of Food Science and Technology (IFST), specified that the maximum permissible levels of yeast and molds are 5 log10 CFU and 4 log10 CFU, respectively [22].

A combination of mathematical and statistical methods, such as the Response Surface Methodology (RSM), is a helpful tool for developing, improving, and optimizing processes [23,24,25,26]. This technique has been employed by numerous studies to improve bakery goods [27,28,29]. Although RSM has been adopted in numerous research as an optimization tool for bakery goods, the optimization of Roselle muffins has not been studied. Therefore, the study’s objective was to formulate and optimize the ingredient levels of the Roselle muffin using RSM. Additionally, the phytochemical analysis of the batter and muffins was investigated to determine their retention after baking.

## 2. Results

### 2.1. Effect of Factors on Muffin Responses

In this study, the effect of Roselle calyx extract, citric acid, and sodium bicarbonate on sensory responses was observed using the Response Surface Methodology (RSM). The average score of 30 responders and the effect of varying factors (extract volume, citric acid, and sodium bicarbonate concentration) on responses (texture and overall acceptability) is presented in Table 1. The texture score ranged from 6.36 to 7.23, while the overall acceptability (OAA) score was between 5.70 and 7.80. Appendix A provides results for other sensory aspects, such as appearance, color, and taste. Figure 1 shows the color of the Roselle muffin when various formulations are combined.

Following up on these observations, analysis of variance (ANOVA) was performed (Table 2). The quadratic model, with an R^2^ value of 0.70 for texture and 0.94 for OAA, was suggested.

This quadratic relationship was observed between the sensory responses and the extract volume, citric acid, and sodium bicarbonate.
Y = 6.96 + 0.06 A − 0.21 B + 0.03 C − 0.10 AB + 0.12 AC + 0.19 BC − 0.27 A^2^ + 0.04 B^2^ − 0.01 C^2^(1)
Y = 7.56 + 0.29 A − 0.11 B + 0.11 C − 0.16 AB + 0.05 AC + 0.24 BC − 0.44 A^2^ − 0.62 B^2^ − 0.01 C^2^(2)

Equation (1) shows the response texture of the optimal quadratic polynomial equation obtained from RSM. Where Y = texture (score), A, B, and C are extract volume, citric acid, and sodium bicarbonate, respectively.

Equation (2) shows the response OAA of the optimal quadratic polynomial equation obtained from RSM. Where Y = OAA (score), A, B, and C are extract volume, citric acid, and sodium bicarbonate, respectively.

Furthermore, the model was determined to be significant for OAA (0.01) and not significant for texture (0.21) using *p* < 0.05 standards. This indicates that the panelists’ preference for OAA was significantly influenced by variations in the Roselle muffin formulation. OAA (0.22) and texture (0.13) were found to have a non-significant Lack of Fit (*p* < 0.05). Non-significant Lack of Fit is favorable because it implies that the response data and the model are compatible [30].

The diagnostic plots of the experiment and model results show the accuracy of the models for Texture (a) and OAA (b) responses (Appendix A).

Given that the points are rationally close to the straight line and no variance was seen, the normal % probability of residual plot for response is normally distributed in both models.

Predicted values are close to the experimental values which shows a good relationship between the actual and the predicted values. In Appendix A, the lowest OAA response value is 5.7 which is indicated by blue; meanwhile, the highest OAA response value is 7.8 which is indicated by red color. This can be interpreted that the actual results would be close to the results predicted by the Design Expert program.

### 2.2. Optimization

Optimization was conducted to obtain responses that are consistent with the responses that the panelists could be perceived. The components and goals in the optimization stage are shown in Table 3. Maximizing the extract volume, texture, and OAA within a range of citric acid and sodium bicarbonate components was the aim of the optimization process.

According to the Design Expert program’s optimization procedure, the following contents were revealed: Citric acid (1.11 g), sodium bicarbonate (1.16 g) and 45.37 mL of the extract volume were present, and the desirability was 0.84 (Table 4). It produces a response score of 8.31 for texture and 8.30 for OAA under such conditions, which is higher than the value predicted by the model (7.13 for texture and 7.27 for OAA). Therefore, it can be stated that optimization and standardization using RSM were successful in the development of the Roselle muffin.

### 2.3. Sensory and Physicochemical Characteristics

The sensory evaluation scores for color and appearance, aroma, body and texture, taste and flavor, and overall acceptability that were received from the semi-trained panelists are presented in Table 5. The muffin’s sensory attributes were valued. Some panelists stated that the Roselle muffin tasted slightly sour, setting it apart from other muffins sold commercially. The high concentration of organic acids, particularly hibiscus acid in Roselle calyx, is what gives muffins their sour flavor [6]. In comparison to the control muffin, the Roselle muffin had higher acceptability scores for attributes, particularly in body and texture (8.21) and taste and flavor (8.18). Similar findings were reported in the study of Siti Faridah et al., which determined that the panelists regarded quality scores with 10% Roselle calyx powder to be the most acceptable [31]. When compared to the control muffin in our study, the muffin enhanced with Roselle calyx extract (not powder) received the highest overall acceptability score (8.3). The addition of Roselle extract enhanced the acceptability. In sensory evaluation, an average liking score of 7 or more on a nine-point scale often indicates extremely acceptable sensory quality [32]. Hence, a product obtaining this score may be taken into consideration as a competent description of the desired quality.

The physicochemical parameters of muffins are presented in Table 5. Roselle calyx extract did not significantly (*p* < 0.05) increase the moisture content of the Roselle muffin. The ash concentration in the Roselle muffin was found to be statistically (*p* < 0.05) higher than that of the control muffin. The high content of ash in the Roselle muffin may be due to the rich amount of calcium, iron, and crude fiber in Roselle calyx. However, compared to the control muffin, the protein and fat content of the Roselle muffin revealed a considerably (*p* < 0.05) lower value. The total carbohydrate content between the Roselle muffin and the control muffin is not significantly different (*p* < 0.05). Comparing the Roselle muffin to the control muffin, the pH of the Roselle muffin is slightly acidic (5.26). Ascorbic acid is present in reasonable amounts in the Roselle muffins (12.1 mg/100 g). For adult women and men, the Dietary Recommended Allowance (RDA) of ascorbic acid is 75 mg per day and 90 mg per day, respectively, to offer antioxidant protection [33].

Color is an important parameter affecting the overall acceptability of the product. Color analysis shows that a Roselle muffin had a reddish color with *L** (62.70 ± 0.19), *a** value (9.51 ± 0.01) and *b** value (10.47 ± 0.06), and that of the control muffin was a brownish-yellow color with *L** (68.84 ± 0.21), *a** value (5.21 ± 0.05), and *b** value (14.88 ± 0.08). In this study, we see that the Roselle muffin had a lighter (*L**) color compared to the control muffin with higher *a** values indicating a red, dark color with a pleasant taste due to the presence of the Roselle calyx extract.

The texture parameters as revealed by instrumental analysis are summarize in Table 5. In every parameter studied, the textural profile of the Roselle muffin showed a significant variation. The Roselle muffin’s texture was significantly changed by sodium bicarbonate and citric acid, giving it a better texture profile than the control muffin. The hardness of the Roselle muffin was reduced by the addition of citric acid and sodium bicarbonate from 1720.84 N in the control to 865.38 N in the Roselle muffin. The volume of the crumb and the overall volume of air cells are connected to hardness [34]. With rising gas cell size, the crumb structure of the product becomes softer. Springiness was observed in the control muffin at 0.89% and the Roselle muffin at 0.87%. There were no obvious changes in the springiness of the control and Roselle muffins. Cohesiveness, which describes how well a food retains its form between the first and second chew, showed a significant difference between the control and Roselle muffins. With the addition of citric acid and sodium bicarbonate, the value of chewiness in the Roselle muffin significantly decreased.

### 2.4. Phytochemical Characteristics of Batter and Optimized Muffin

The phytochemical attributes of batters and Roselle muffins are presented in Table 6. In total, the batter contained 154.56 mg C3G of anthocyanins (TAC/100 g). The TAC was 126.63 mg Cyn-3-glu/100 g after baking; which suggests that about 82% of the TAC was retained. The total phenolic content (TPC) in the batter for the Roselle muffin significantly decreased from 19.74 to 12.91 mg GAE/100 g. Indicating that only 65% of the TPC was retained. The antioxidant activity of the batter was 27.30%, while that of the Roselle muffin was 12.53%. Thus, the reduction in antioxidant activity was nearly 50%. There was a modest reduction in TAC and antioxidant properties after baking.

### 2.5. Microbial Shelf-Life

In this study, the muffins’ microbiological shelf-life was examined every three days and are shown in Table 7. In the first and third days of room-temperature storage (25 °C), the overall plate counts were below the limit for detection. The sixth day of storage saw the appearance of a few bacterial colonies in the Roselle muffin. After the sixth day of storage, there were an infinite number of colonies. The analysis was stopped on day 12 based on the observation of microbial growth, which showed that molds had begun to form on the ninth day of storage in the Roselle muffin. According to this study, during the ninth day of storage at room temperature, both the total plate count and the yeast and mold count increased in the Roselle muffin. However, the control muffin is within the permissible limit up to day 9 of the storage. This may be due to the presence of anthocyanins in Roselle calyx extract. Microencapsulation may be the solution for improving the stability of anthocyanins [35]. Aerobic plate counts acceptable values ranged from 5 log10 CFU to 6 log10 CFU [21] and maximum permissible levels of yeast and molds 5 log10 CFU 4 log10 CFU, respectively [22]. Because the results of the microbiological count were higher than the permissible limit, the findings of the present study suggest that Roselle muffins may not be safe to consume after the sixth day of storage at room temperature. It is mostly due to no preservatives added to the muffin. Without preservatives, these microorganisms will eventually degrade baking products, reducing their shelf life. Our study is in line with that of Rodriguez et al., who discovered that cake samples with no preservative were moldy on the surface after 6 days of storage at 15–20 °C [36].

## 3. Materials and Methods

### 3.1. Chemicals

Trichloroacetic acid (99%), Gallic acid (98%), and 2,2-diphenyl-1-picrylhydrazyl (DPPH, 95%) were obtained from Sisco Research Laboratory Pvt. Ltd. (Mumbai, India). L-ascorbic acid (99%); sodium carbonate, anhydrous (99.5%); glacial acetic acid (99.5%); and petroleum ether were purchased from Thermo Fisher Scientific (Mumbai, India). 2,6-dichlorophenol-indophenol sodium salt (DCIP) and the Folin & Ciocalteus Phenol (FCP) reagent were obtained from Central Drug House (P) Limited (New Delhi, India); nutrient agar and potato dextrose agar (PDA) were obtained from HiMedia; methanol and ethanol of analytical grade were procured from Rankhem (New Delhi, India).

### 3.2. Raw Materials

The research material was centered on the development of nutritionally enriched muffins. In the current study, dried Roselle (*H. sabdariffa*) calyces from the Garo Hills in Meghalaya, India, were used as a source of natural pigment [6]. Flour, sugar, butter, eggs, salt, citric acid powder, and sodium bicarbonate have also been acquired at the neighborhood store in order to carry out the muffin-making process.

### 3.3. Design of the Experiment

The experiment was planned using the Response Surface Methodology (RSM) Box–Behnken’s Design (BBD). Operationally, the method was designed through data processing by applying the Design Expert (DX) Program Version 13.0.6, Stat-Ease Inc., Minneapolis, MN, USA (www.statease.com (accessed on 23 August 2021)). Three independent factors (variables), namely, extract volume, citric acid, and sodium bicarbonate, were employed in this study; their respective codes were A, B, and C. The findings of the preliminary study were used to determine the independent variables. The independent variables’ minimum and maximum values were entered into the model for randomization. Table 8 displays every level of the variable included in the model. For the three independent variables, a total of 17 runs were generated after randomization (Table 9).

### 3.4. Roselle Calyx Extract Preparation

For extraction, the fine calyx powder (10 g) was heated for 5 min at 80° C after soaking overnight at room temperature in 100 mL of distilled water. The extract was filtered through filter paper and made the final volume to 100 mL. It was then added to the batter of the muffins.

### 3.5. Muffin Preparation

The control muffin was based on a simple vanilla muffin recipe [37] with a slight modification (Table 9) while in the treatments, the baking powder was replaced with citric acid and sodium bicarbonate to maintain the pH of the batter as the color of anthocyanin is pH dependent, and the Roselle extract was added in place of milk in the proportion mentioned in Table 9. It involved combining all the specified ingredients in an electric mixer and blending them at high speed for five minutes. The batter was combined, then poured into a muffin pan, and baked for 20 min at 180 °C. (OTG Wonder Chef oven, Wonder Chef, Mumbai, India). Five batches were prepared for the optimized formulation. The standard vanilla muffin recipe was used to make the control muffin. The muffins were packed in a clean metallic polyethylene zip lock bag (32-micron) after baking and cooling in preparation for further testing.

### 3.6. Optimization

The Design Expert Program’s optimization objective was used to evaluate each response (texture and overall acceptability). According to the data from the fitted variables and the response score, optimization was carried out. The Design Expert Program interprets the results of the optimization and generate new optimum formulation.

### 3.7. Sensory Analysis

The muffins’ sensory preferences were assessed using a 9-point hedonic scale, with 1 being highly dislike and 9 representing a favorable opinion (like very much). The samples, which were distributed at random, were analyzed by a semi-trained panel comprising 30 panelists. All the panelists were familiar with food tasting and capable of differentiating the taste, flavor, texture, etc., and presenting their reactions. The panelists ranged in age from 15 to 60. The presentation of the samples was carried out in a random order, and the sample sets were allotted with different codes. Each time on the day of analysis, the panelists were given four sets of different formulations and one control sample for sensory evaluation. Each formulation was tested by 30 panelists (*n* = 30). Taste-neutral water was provided for rinsing, and the samples were presented at room temperature.

### 3.8. Physicochemical Analysis

For all analyses, three replicates were taken and each replicate consisted of one muffin. The details of the methodology are given below.

#### 3.8.1. Proximate

The proximate composition of the optimized and control muffins in terms of moisture, fat, protein, ash, and total carbohydrates were determined using standardized and validated laboratory AOAC methods [38].

#### 3.8.2. Texture Profile Analysis

The method provided by Jauraha et al. served as the basis for the instrumental analysis of muffin texture [39]. Using the TA-XT-plus Texture Analyzer, a texture profile analysis of the optimized and control muffins was performed at IARI, PUSA, New Delhi, India. For compression, a 75 mm diameter P/35” aluminum plate was employed. The following conditions were used to conduct the test: Test speed was 1 mm/s, the trigger force was 5 g, and the strain was 50%. The muffin was compressed twice to assess its textural characteristics, such as its hardness, springiness, cohesiveness, and chewiness.

#### 3.8.3. PH Measurement

A pH meter (LMPH-10, LABMAN Scientific instrument, Chennai, India) was used to measure the muffin’s pH [40]. To separate the solids and liquid, 0.5 g of the sample was treated with 20 mL of distilled water, vortexed for 3 min, and then, left at room temperature for 1 h. The pH of the supernatant was determined after centrifugation for 3 min at 3050× *g*.

#### 3.8.4. Color Measurement

Using a CIE color measuring instrument (NR100 Precision colorimeter, 3nh, Shenzhen, China) and the *L** *a** *b** color scale system, the measurement of color was analyzed [40]. The *a** value stands for redness/greenness, *b** value for yellowness/blueness, and *L** value for lightness/darkness. For color analysis, 1 muffin was powdered, of which 20 g of a muffin was pulverized into small particles prior to the examination and placed on a particular plate for analysis. The muffin’s color was automatically detected and displayed on the screen.

### 3.9. Phythochemical Analysis

#### 3.9.1. Total Phenolic Content

The Folin–Ciocalteu method was used to determine the total phenolic content (TPC) [41]. In brief, a stock solution of gallic acid (1 mg/mL) was prepared at different concentrations in methanol to produce a standard gallic acid curve. These concentrations were mixed with 2 mL of 10% Folin–Ciocalteu reagent. Then, 1 mL of 10% sodium carbonate solution was added after 6 min. After 90 min, the absorbance was measured at 760 nm using UV–VIS spectrophotometer (LMSP-UV1900, LABMAN). For the sample extracts, the same process was repeated. Gallic acid equivalents (mg GAE/g) were used to calculate the total amount of phenol in the sample.

#### 3.9.2. Total Anthocyanin Content

According to the method provided by Lee et al., the total anthocyanin content (TAC) was determined [42]. A total of 10 mL of distilled water were used to dilute 1 mL of the sample extract. Then, 1 mL of the diluted sample solution was diluted with 5 mL with buffer pH 1.0 into a test tube and wrapped with aluminum foil. next, 1 mL of the diluted sample solution was further diluted with 5 mL with buffer pH 4.5 into test tube and wrapped with aluminum foil. The absorbance at 520 nm and 700 nm was measured using a UV–VIS spectrophotometer (LMSP-UV1900, LABMAN) in a 4 mL spectrophotometer glass cell after the mixtures were let to stand for 30 min at room temperature. According to the following equation, results were reported as Cyanidin-3-glucoside equivalents per 100 g of sample.
CA = (A × MW × DF × 100)/(ε × 1)

CA: Anthocyanin concentration (mg/100 g); A: absorbance difference A (A520 nm–A700 nm) pH 1.0 − (A520 nm–A700 nm) (A520 nm–A700 nm) pH 4.5; MW (Molecular Weight): cyanidin-3-glucoside weighs 449.2 g/mol; DF: dilution factor; 1: pathlength in cm; ε: cyanidin-3-glucoside’s molar extinction co-efficient (26,900); 100: conversion factor for deriving mg/100 g.

#### 3.9.3. DPPH Radical Scavenging

The procedure described by Brand-Williams et al. [43] with a slight modification was used to carry out the DPPH radical scavenging activity. In brief, 5.92 mg of DPPH was dissolved in 100 mL of methanol to make a DPPH solution. Then, 1 mL sample extract and 3 mL of DPPH were mixed, vortexed, and kept in the dark at room temperature for 30 min. The control sample, without any extract, was prepared in the same volume. Then, using methanol as a blank, the absorbance was spectrophotometrically measured at 515 nm using UV–VIS spectrophotometer (LMP-UV1900, LABMAN). The following equation was used to determine the radical scavenging activity
% inhibition = (Abs of control − Abs of sample/Abs of control) × 100

#### 3.9.4. Ascorbic Acid Content

Ascorbic acid content was determined using 2,6-dichlorophenolindophenol (DCIP) dye-titrimetric method [40]. The TCA solution was prepared by dissolving 15 g of trichloroacetic acid (TCA) in 200 mL of distilled water and 40 mL of acetic acid. Thereafter, it was diluted with distilled water to 500 mL and filtered using filter paper. To make a standard ascorbic acid solution, weigh 0.05 g ascorbic acid and dissolve it in 60 mL TCA solution and make a final volume of 250 mL with distilled water. The dye standard solution was prepared by dissolving 0.05 g of 2,6-dichlorophenolindophenol and 0.05 g sodium carbonate (Na_2_CO_3_) in 100 mL distilled water. The dye was standardized by titrating it against 10 mL of ascorbic acid stock solution until a faint pink color that lasted for a few seconds was obtained

In order to prepare the sample, 0.5 g was weighed and 20 mL of TCA solution was added to it. The mixture was diluted to the desired strength with distilled water and filtered into a volumetric flask. An aliquot of 10 mL was titrated against the reference indophenol solution. The ascorbic acid was estimated as
Ascorbic acid (mg/100 g) = C × V × DF/W × 100
where C: mg ascorbic acid/mL dye; V: volume of aliquot used for titration; DF: dilution factor; W: weight of sample.

### 3.10. Microbial Shelf-Life

After baking and cooling, muffins were packed in metallic polyethylene zip lock bag (32-micron) and stored at room temperature (25 °C). The muffins were examined for microbial growth at intervals of 3 days. The microbial shelf life was carried out using the plate count method to determine the total plate counts using Nutrient Agar (HiMedia, Mumbai, India) and yeasts and molds using Potato Dextrose Agar (HiMedia). Potato Dextrose Agar plates were incubated at 25 °C for 48 h and Nutrient Agar plates were incubated at 37 °C for 24 h and the results were expressed as colony-forming units per gram (cfu/g).

### 3.11. Statistical Analysis

The experiment was statistically analyzed using the Box–Behnken Design based on the RSM. Quadratic ANOVA was used to assess each response variable. ANOVA was used to examine the influence of different variables on the outcome. For comparing the means between the control and the optimized muffin, ANOVA using excel at *p*-value 0.05 significance level was used.

## 4. Conclusions

The use of the Response Surface Methodology allowed us to optimize the Roselle muffin and achieved significant enhancement of its quality parameters: consumer acceptability, nutritional properties, and shelf-life. Using the BBD model of RSM it was possible to predict the quality parameters of a new type of muffin. The outcome of the study showed that adding Roselle calyx extract to muffins increases their nutritional value by obtaining better antioxidants, bioactive components, total phenolic, and vitamin c that may have the potential of having many health benefits. Microbial shelf-life indicated that Roselle muffin can be stored for 6 days at room temperature with no preservatives added.

## Figures and Tables

**Figure 1 foods-11-03982-f001:**
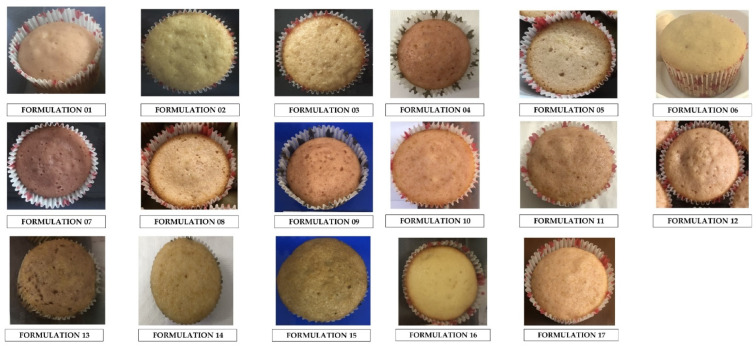
Images of Roselle muffin obtained with different formulations combinations.

**Table 1 foods-11-03982-t001:** Processed data on the muffin score given by the respondents.

	Independent Variables			Dependent Variables			
Codes	A	B	C	Response 1: Texture (Score)		Response 2: OAA (Score)	
Formulation	Extract Volume (mL)	Citric Acid (g)	Sodium Bicarbonate (g)	Experimental Data	RSM Predicted Data	Experimental Data	RSM Predicted Data
01	27.5	2.5	1.0	6.36 ± 1.54	6.54	5.70 ± 1.78	5.88
02	5.0	1.0	1.7	6.63 ± 0.92	6.90	6.00 ± 1.11	6.14
03	27.5	1.7	1.7	6.93 ± 1.11	6.96	7.53 ± 0.97	7.56
04	50.0	2.5	1.7	6.60 ± 1.28	6.47	6.65 ± 1.25	6.50
05	27.5	1.0	1.0	7.23 ± 0.97	6.96	6.53 ± 1.27	6.35
06	5.0	1.7	2.5	6.46 ± 1.00	6.51	6.26 ± 0.94	6.30
07	50.0	1.7	2.5	6.63 ± 1.35	6.58	6.78 ± 0.99	6.99
08	27.5	2.5	2.5	7.13 ± 1.27	7.00	6.66 ± 1.24	6.59
09	50.0	1.7	1.0	6.63 ± 1.09	6.76	6.70 ± 1.08	6.66
10	27.5	1.7	1.7	6.88 ± 0.63	6.96	7.30 ± 0.83	7.56
11	27.5	1.7	1.7	7.23 ± 0.72	7.05	7.80 ± 0.80	7.56
12	27.5	1.7	1.7	7.00 ± 0.69	6.96	7.60 ± 0.77	7.56
13	50.0	1.0	1.7	7.20 ± 1.03	7.12	7.10 ± 0.99	7.07
14	5.0	1.7	1.0	6.96 ± 1.15	6.69	6.38 ± 1.32	6.17
15	27.5	1.0	2.5	7.23 ± 0.85	7.36	6.53 ± 0.93	6.60
16	5.0	2.5	1.7	6.46 ± 1.16	6.54	6.20 ± 1.12	6.23
17	27.5	1.7	1.7	6.76 ± 0.67	6.96	7.56 ± 0.62	7.56

**Table 2 foods-11-03982-t002:** Result of analysis of variance (ANOVA) texture response (A) and OAA response (B).

A						B					
Source	Sum of Squares	Degree of Freedom	Mean Square	F-Value	*p*-Value	Source	Sum of Squares	Degree of Freedom	Mean Square	F-Value	*p*-Value
Model	1.01	9	0.1125	1.86	0.21 **	Model	5.62	9	0.6244	12.82	<0.01 *
A-Extract Volume	0.0378	1	0.0378	0.62	0.45	A-Extract Volume	0.7140	1	0.7140	14.66	<0.01
B-Citric Acid	0.3784	1	0.3784	6.26	0.04	B-Citric Acid	0.1128	1	0.1128	2.32	0.17
C-Sodium Bicarbonate	0.0091	1	0.0091	0.15	0.70	C-Sodium Bicarbonate	0.1058	1	0.1058	2.17	0.18
AB	0.0462	1	0.0462	0.76	0.41	AB	0.1056	1	0.1056	2.17	0.18
AC	0.0625	1	0.0625	1.03	0.34	AC	0.0100	1	0.0100	0.20	0.66
BC	0.1482	1	0.1482	2.45	0.16	BC	0.2304	1	0.2304	4.73	0.06
A²	0.3242	1	0.3242	5.37	0.05	A²	0.8441	1	0.8441	17.33	<0.01
B²	0.0067	1	0.0067	0.11	0.74	B²	1.63	1	1.63	33.52	<0.01
C²	0.0007	1	0.0007	0.01	0.91	C²	1.42	1	1.42	29.10	<0.01
Residual	0.4230	7	0.0604			Residual	0.3410	7	0.0487		
Lack of Fit	0.3012	3	0.1004	3.30	0.13	Lack of Fit	0.2133	3	0.0711	2.23	0.22
Pure Error	0.1218	4	0.0305			Pure Error	0.1277	4	0.0319		
Cor Total	1.44	16				Cor Total	5.96	16			
Fit Statistics	Std. Dev.	Mean	C.V%	R^2^		Fit Statistics	Std. Dev.	Mean	C.V%	R^2^	
	0.2458	6.84	3.59	0.70			0.2207	6.78	3.25	0.94	

OAA: Overall acceptability analysis; * significant; ** non-significant.

**Table 3 foods-11-03982-t003:** Components and optimization goal.

Components	Goal	Lower Limit	Upper Limit
A: Extract Volume	Maximize	5.00	50.0
B: Citric Acid	is in range	1.00	2.50
C: Sodium Bicarbonate	is in range	1.00	2.50
Texture	Maximize	6.36	7.23
Overall Acceptability	Maximize	5.70	7.80

**Table 4 foods-11-03982-t004:** The optimization and outcome of the optimized responses.

**Components**	**Optimum Value**	
Extract Volume (mL)	45.37	
Citric acid (g)	1.11	
Sodium bicarbonate (g)	1.67	
Desirability of response value	0.84	
**Response**	**Model Calculated value**	**Observed value**
Texture	7.13	8.31
OAA	7.27	8.30

**Table 5 foods-11-03982-t005:** Sensory and physicochemical characteristics of muffin.

Parameters	Roselle Muffin	Control Muffin	*p*-Value (*p* < 0.05)
**Sensory Parameters**			
Colour and Appearance	7.85 ± 0.70 *	7.36 ± 0.80 *	0.01
Aroma	7.71 ± 0.66 ^ns^	7.70 ± 0.70 ^ns^	0.92
Body and Texture	8.21 ± 0.53 **	7.53 ± 0.89 **	<<0.01
Taste and Flavour	8.18 ± 0.70 *	7.76 ± 0.81 *	0.03
Overall Acceptability	8.30 ± 0.68 **	7.68 ± 0.64 **	<<0.01
**Proximate** (per 100 g)			
Moisture (%)	21.66 ± 0.89 *	20.86 ± 0.24 *	0.29
Ash (%)	4.22 ± 0.41 **	2.11 ± 0.41 **	<<0.01
Fat (%)	11.20 ± 0.16 **	13.6 ± 0.16 **	<<0.01
Protein (%)	7.73 ± 0.12 **	8.33 ± 0.12 **	<<0.01
Total Carbohydrate (%)	55.18 ± 1.24 ^ns^	55.09 ± 0.51 ^ns^	0.92
pH	5.26 ± 0.04 **	7.10 ± 0.08 **	<<0.01
Ascorbic acid, mg/100 g	12.10 ± 0.89	ND	-
**Color Parameters**			
*L**	62.70 ± 0.19 **	68.84 ± 0.21 **	<<0.01
*a**	9.51 ± 0.01 **	5.21 ± 0.05 **	<<0.01
*b**	10.47 ± 0.06 **	14.88 ± 0.08 **	<<0.01
*c**	14.14 ± 0.05 **	15.77 ± 0.10 **	<<0.01
*h* ^0^	47.75 ± 0.11 **	70.69 ± 0.06 **	<<0.01
**Texture Parameters**			
Hardness, N	865.38 ± 6.32 **	1720.84 ± 56.08 **	<<0.01
Springiness, %	0.87 ± 0.01 ^ns^	0.89 ± 0.00 ^ns^	0.30
Cohesiveness	0.45 ± 0.00 *	0.50 ± 0.01 *	0.02
Chewiness	346.26 ± 1.71 **	780.11 ± 41.87 **	<<0.01

ND = not detected; * significant; ** highly significant; ^ns^ non-significant.

**Table 6 foods-11-03982-t006:** Phytochemical content of batter and muffin.

	Roselle Muffin		Control Muffin		
Phytochemical Parameters	Batter	Muffin	Batter	Muffin	*p*-Value (*p* < 0.05)
TAC, mg cyanidin-3-glucoside (Cyn-3-glu/100 g)	154.56 ± 3.40 **	126.63 ± 1.96 **	ND	ND	<<0.01
TPC, mg gallic acid (GA/100 g)	19.74 ± 0.28 **	12.91 ± 0.69 **	3.12 ± 0.06	ND	<<0.01
Antioxidant activity (%)	27.30 ± 1.03 **	12.53 ± 0.13 **	ND	ND	<<0.01

TAC: Total anthocyanin content; TPC: Total phenolic content; GA: Gallic acid; ND: Not Detected; ** highly significant.

**Table 7 foods-11-03982-t007:** Total plate count and yeast and molds during storage of muffins (cfu/g).

	**Roselle Muffin**		**Control Muffin**	
Days	Total Plate Count	Yeast and Molds	Total Plate Count	Yeast and Molds
01	2.4 × 10^3^	4.1 × 10^3^	3 × 10^2^	2 × 10^2^
03	1.8 × 10^4^	1.5 × 10^4^	3 × 10^2^	5 × 10^2^
06	5.4 × 10^4^	4.9 × 10^4^	7 × 10^3^	3 × 10^3^
09	Uncountable	Uncountable	5.7 × 10^4^	4.5 × 10^4^
12	Visible mold growth	Visible mold growth	Uncountable	Uncountable
15	Visible mold growth	Visible mold growth	Visible mold growth	Visible mold growth

**Table 8 foods-11-03982-t008:** Variable levels for preparing muffins.

Codes	Independent Variables	Units	Minimum	Maximum
A	Extract Volume	mL	5.00	50.00
B	Citric Acid	g	1.0000	2.50
C	Sodium Bicarbonate	g	1.0000	2.50

**Table 9 foods-11-03982-t009:** Experimental design for preparation of Roselle muffins based on RSM along with control.

Independent Variables					Dependent Variables					
Formulation	A: Extract Volume (mL)	B: Citric Acid (g)	C: Sodium Bicarbonate (g)	Milk Volume (mL)	Flour (g)	Sugar (g)	Butter (g)	Egg * (g)	Vanilla Extract (g)	Salt (g)
1	27.5	2.5	1.0	0	100	60	50	100	1	1
2	5.0	1.0	1.7	0	100	60	50	100	1	1
3	27.5	1.7	1.7	0	100	60	50	100	1	1
4	50.0	2.5	1.7	0	100	60	50	100	1	1
5	27.5	1.0	1.0	0	100	60	50	100	1	1
6	5.0	1.7	2.5	0	100	60	50	100	1	1
7	50.0	1.7	2.5	0	100	60	50	100	1	1
8	27.5	2.5	2.5	0	100	60	50	100	1	1
9	50.0	1.7	1.0	0	100	60	50	100	1	1
10	27.5	1.7	1.7	0	100	60	50	100	1	1
11	27.5	1.7	1.7	0	100	60	50	100	1	1
12	27.5	1.7	1.7	0	100	60	50	100	1	1
13	50.0	1.0	1.7	0	100	60	50	100	1	1
14	5.0	1.7	1.0	0	100	60	50	100	1	1
15	27.5	1.0	2.5	0	100	60	50	100	1	1
16	5.0	2.5	1.7	0	100	60	50	100	1	1
17	27.5	1.7	1.7	0	100	60	50	100	1	1
Control	0	0	0	50	100	60	50	100	1	1

* Whole raw beaten egg at room temperature.

## Data Availability

The data presented in this study are available in Appendix A.

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
