# Peer review of "Nutritionally Enriched Muffins from Roselle Calyx Extract Using Response Surface Methodology"

_foods, 2022, doi:10.3390/foods11243982_

Round 1

Reviewer 1 Report

Line 53  Delete 'to offering basic nutrition"

Table 1  Should have standard deviations and stats letters for Tukey's test if ANOVA was significant, so you can compare the formulations to each other.

Table 2 Define OAA.  Figure 2 is not really needed.

line 111 Delete 'variation'.

line 114 to 115 Please clarify what you mean by close.

Table 5 What statistical analysis was used for the p-value result?  Was a t-test used?  If so, indicate this in the methods.

line 194  Did you mean to say 60%?

line 257 to 264  Please state how many batches of each formulation were made.  Also provide the percentages of all ingredients used for the sample and also for the control.  You need a reference for the 'standard vanilla muffin recipe'.

line 271 to 277  You need to provide details about how panelists were trained and how many samples each panelist tasted and how many times each formulation was tested by the panelists.

line 288 and 289  Use 'was' not 'is'.

For all analyses state how many muffins of each formulation were used from each batch.

line 300 How do you know you had a representative sample?  What part of the muffin did your take the 20 grams from?  It would have been more informative to do both crust and crumb color separately.

line 320 to 321  Why was this anthocyanin chosen as the equivalent?

line 362 to 364  You should use Tukey's test to compare means between formulations, also a t-test for the control versus ideal formulation sample muffin.

Reviewer 2 Report

Nutritionally Enriched Muffins from Roselle Calyx Extract using Response Surface Methodology

·       What is the reason for using Response Surface Methodology (RSM) in this study?

·       In the introduction, the anthocyanins decay very quickly and the color changes very fastly, how to solve this problem?

·       In the case of muffin response (the overall acceptability, OAA), was it compared with the control group?

·       Specify the origin of the concentration ranges of citric acid and sodium bicarbonate used in the formulation (Table 3).

·       What are the types of packaging used for stability studies?

·       In case of microbiology testing, acceptance criteria should be specified. P. aeruginosa, E. coli, S. aureus, Enterobacteriaceae, toxin, and insecticide in Roselle muffin should be performed.

·         Stability studies should be shown under various conditions, including color changes. and various important substances.

·       The study how to increase the stability of Roselle muffin should be done. Encapsultion of anthocyanins may increase the stability.

·      Add information of ethics

Reviewer 3 Report

Recommendation: Major

The manuscript Nutritionally Enriched Muffins from Roselle Calyx Extract using Response Surface Methodology, the methodology was reasonable and technically sound.

Comments to the Author:

The main procedure and findings of the study are well expressed. Introduction: A brief survey of existing literature, the purpose, importance, and innovation of the research is well mentioned. The tables and graphs are well prepared.

Below are some important suggestions.

Point 1. In the abstract, include not only the data of the optimization results but also the numerical results about the increase in nutrients.

Point. Line 76-77 A combination of mathematical and statistical methods, such as the Response Surface  Methodology (RSM), is a helpful tool for developing, improving, and optimizing processes (add citation).  I would suggest you add up-to-date RSM studies below.

https://doi.org/10.3390/molecules27217395

https://doi.org/10.3390/pr10102100

https://doi.org/10.3390/molecules27165222

https://doi.org/10.3390/foods11172709

Point 3. Line 81, Table S1 ?

Point 4. For Figure 1, set the resolution to at least 300 dpi. also match the codes of the products with table 1.

Point 5. For Table 1, add the RSM results next to the experimental results.

Point 6. Add the polynomial equations of the overall evaluation and texture results to the article.

Point 7. With which method did you statistically compare the results of Table 5 and Table 6 ? Add it to the statistical analysis section.

Point 8. Provide details of chemicals and devices in journal format

Point 9. Explain the RSM design in detail by writing the dependent and independent variables.

Point 10. Why was the total color change not taken into account in the color measurement? I recommend adding it if possible.

Point 11. Table 5 ascorbic acid unit in mg? mg/100g

Point 12. Line 344 chemical Na2CO3 names should be written correctly.

Point 13. Write the company details of the media used in microbiological analysis. Also, write in which unit the results are expressed.

Point 14. Line 236 (26,900) dot comma?

Point 15. Why battery in phytochemical analysis?

Round 2

Reviewer 2 Report

All the equations are in a different format. Please rearrange it in the journal guidelines.

2.5. Microbial shelf-life; the humidity should be present in this section.

Reviewer 3 Report

The authors have made the necessary revisions for the article.